# CORK1, A LRR-Malectin Receptor Kinase, Is Required for Cellooligomer-Induced Responses in *Arabidopsis thaliana*

**DOI:** 10.3390/cells11192960

**Published:** 2022-09-22

**Authors:** Yu-Heng Tseng, Sandra S. Scholz, Judith Fliegmann, Thomas Krüger, Akanksha Gandhi, Alexandra C. U. Furch, Olaf Kniemeyer, Axel A. Brakhage, Ralf Oelmüller

**Affiliations:** 1Matthias Schleiden Institute of Genetics, Bioinformatics and Molecular Botany, Department of Plant Physiology, Friedrich-Schiller-University Jena, 07743 Jena, Germany; 2Center for Plant Molecular Biology (ZMBP), University of Tübingen, 72074 Tübingen, Germany; 3Department of Molecular and Applied Microbiology, Leibniz Institute for Natural Product Research and Infection Biology-Hans Knöll Institute (Leibniz-HKI), 07745 Jena, Germany; 4Department of Microbiology and Molecular Biology, Institute of Microbiology, Friedrich Schiller University, 07743 Jena, Germany

**Keywords:** cellooligomer, cellotriose, cellulose, LRR receptor kinase, malectin, cell wall integrity

## Abstract

Cell wall integrity (CWI) maintenance is central for plant cells. Mechanical and chemical distortions, pH changes, and breakdown products of cell wall polysaccharides activate plasma membrane-localized receptors and induce appropriate downstream responses. Microbial interactions alter or destroy the structure of the plant cell wall, connecting CWI maintenance to immune responses. Cellulose is the major polysaccharide in the primary and secondary cell wall. Its breakdown generates short-chain cellooligomers that induce Ca^2+^-dependent CWI responses. We show that these responses require the malectin domain-containing CELLOOLIGOMER-RECEPTOR KINASE 1 (CORK1) in *Arabidopsis* and are preferentially activated by cellotriose (CT). CORK1 is required for cellooligomer-induced cytoplasmic Ca^2+^ elevation, reactive oxygen species (ROS) production, mitogen-associated protein kinase (MAPK) activation, cellulose synthase phosphorylation, and the regulation of CWI-related genes, including those involved in biosynthesis of cell wall material, secondary metabolites and tryptophan. Phosphoproteome analyses identified early targets involved in signaling, cellulose synthesis, the endoplasmic reticulum/Golgi secretory pathway, cell wall repair and immune responses. Two conserved phenylalanine residues in the malectin domain are crucial for CORK1 function. We propose that CORK1 is required for CWI and immune responses activated by cellulose breakdown products.

## 1. Introduction

The primary cell wall is mainly composed of polysaccharide polymers, including cellulose, hemicelluloses and pectins. Cellulose accounts for more than 30% of the primary cell wall material [1] and consists of β-(1,4)-bound D-glucose moieties, which form unbranched fibers with a paracrystalline structure. Hemicelluloses are made of xylans, xyloglucans, mannans, glucomannans and β-(1,3;1-4)-glucans. The backbone for xylans, xyloglucans and mannans is made of β-(1,4)-linked monomer residues, while β-(1,3;1-4)-glucans contain β-(1-4)-linked glucose interleaved with β-(1,3) linkages. Unlike cellulose, hemicelluloses have short branches, and their amorphous structure is easily accessible to hydrolases [2,3]. Pectins are categorized into unbranched homogalacturonan (HG) and branching rhamnose (Rha)-containing rhamnogalacturonan I (RG-I) and rhamnogalacturonan II (RG-II) with complex composition [4,5]. While HG is a monopolymer of α-(1,4) galacturonic acid (GalA), RG-I has a disaccharide unit backbone of α-D-GalA-(1,2)- α-L-Rha, and RG-II possesses GalA linked with various sugars [6,7].

Fragments of these cell wall polysaccharides have been shown to act as damage-associated molecular patterns (DAMPs) [8]. Cellulose breakdown products, cellooligomers (COMs), trigger calcium influx, ROS production, mitogen-activated protein kinase (MAPK) activation and defense-related gene expression, which eventually leads to higher pathogen resistance [9,10,11,12,13,14,15]. COMs with 2-7 glucose moieties induce cytoplasmic calcium ([Ca^2+^]_cyt_) elevation. The amplitude of the response depends on the length of the oligomer, and cellotriose (CT) has been found to be the most active COM in *Arabidopsis thaliana* [13]. The defense responses induced by COMs are relatively mild when compared to those induced by the pathogen-associated molecular patterns (PAMPs) chitin or flg22 [11,12,13,16]. However, in combination with chitin, flg22 or oligogalacturonic acid (OG), synergistic effects to calcium influx, ROS production and MAPK activation indicate crosstalk between COM and PAMP responses [11,13].

Plants rely on an array of membrane-associated pattern recognition receptors (PRRs) to recognize breakdown products of its cell wall. The wall-associated kinase 1 (WAK1) is activated by the pectin fragment OGs [17,18]. FERONIA, a membrane-localized receptor-like kinase with a malectin-like domain, is involved in monitoring cell wall integrity (CWI), pollen tube development, plant growth and perception of rapid alkalinization factor (RALF) peptides [19,20]. Its extracellular region has been shown to interact with pectin [21,22]. In rice, two species of mixed-linked β-1,3/1,4-glucans (MLGs) from hemicellulose, namely 3^1^-β-D-cellobiosyl-glucose and 3^1^-β-D-cellotriosyl-glucose, bind to OsCERK1 and induce the dimerization of OsCERK1 and the chitin receptor OsCEBiP [15]. However, up to this point, no receptor has been reported to perceive β-1,4 glucans. In this study, we show that CORK1 (cellooligomer receptor kinase 1) is vital in COM-induced physiological responses in *A. thaliana*. CORK1 is a functional LRR-malectin receptor kinase. Upon COM treatment, *cork1* mutants are impaired in [Ca^2+^]_cyt_ elevation, ROS production and regulation of genes involved in CWI maintenance and immune responses, including *WRKY30*/*WRKY40*. Two conserved phenylalanine residues in the malectin domain are crucial for COM-induced responses in *A. thaliana*. Phosphoproteome and transcriptome data identify putative protein and gene targets of the novel COM/CORK1 pathway and shed light on the role of COMs in CWI maintenance.

## 2. Materials and Methods

### 2.1. Growth Medium and Conditions for Seedlings

*A. thaliana* seeds were surface-sterilized for 8 min in sterilization solution containing lauryl sarcosine (1%) and Clorix cleaner (23%). Surface-sterilized seeds were washed with sterilized water 8 times and placed on Petri dishes with MS medium supplemented with 0.3% gelrite [23]. After cold treatment at 4 °C for 48 h, plates were incubated at 22 °C under long day conditions (16 h light/8 h dark; 80 μmol m^−2^ s^−1^).

Wild-type (ecotype Columbia-0), the aequorin-containing wild-type [pMAQ2] line (AeqWT) [24] and EMS (ethyl methanesulfonate)-induced mutant lines of AeqWT background [13] were used in this study. In addition, two T-DNA insertion lines, *cork1-1* (N671776; SALK_099436C) and *cork1-2* (N674063; SALK_021490C), were obtained from Nottingham *Arabidopsis* Stock Centre (NASC). Homozygous seedlings of these insertion lines were crossed to the AeqWT. The corresponding segregated wild-type (SWT) and homozygous (HO) seedlings from the F3 generation were used for experiments.

### 2.2. EMS Mutagenesis of A. thaliana Seeds

Two-and-a-half g of AeqWT seeds were used for mutagenesis. According to Kim et al. [25], seeds were soaked in 40 mL of 100 mM phosphate buffer (pH = 7.5) for 10 h at 4 °C. The next day, the buffer was replaced, and EMS (Sigma-Aldrich, Taufkirchen, Germany) was added to a final concentration of 0.2%. The mixture was incubated at room temperature in a hood overnight with gentle stirring. The seeds were washed twice in 40 mL of 100 mM sodium thiosulphate for 15 min to destroy the remaining EMS, followed by 18 wash steps with water [26]. Freshly mutagenized seeds were directly separated in different Eppendorf tubes, surface-sterilized and germinated as described above. Three-week-old plants were transferred to soil to obtain the seeds of the individual mother plants.

### 2.3. Whole Genome Sequencing and SNP Analysis

After screening ~100 independent EMS lines, we found a COM non-responsive mutant, named here as EMS71. From the F2 generation of the back-cross between EMS71 and AeqWT, two pools of seedlings were sorted out. One pool consisted of CT-responsive individuals, while the other contained non-responsive individuals (50 seedlings in each pool). Whole-genome sequencing of the two pools was performed on an Illumina sequencing platform (PE150; Novogene Co., Cambridge, UK). The reads from both pools were mapped separately against the TAIR10 reference genome using Samtools v1.8 [27]. SIMPLE v1.8.1 [28] analysis was implemented to filter out single-nucleotide polymorphism (SNP) with a frequency of 90–100% in the non-responsive population, while less than 70% in the responsive population. Putative candidate genes were selected based on whether the SNP caused non-synonymous mutation or affected mRNA processing (e.g., mRNA splicing). SNP sites of candidate genes were confirmed in three different individuals from the EMS71 line. As a result, two candidate genes, *ARF1* (*AT1G59750*) and *CORK1* (*AT1G56145*), were selected for further confirmation as described in the Section 3.

### 2.4. Transcriptome Analysis

Sixteen roots of SWT or HO from the *cork1-2* mutant line crossed to AeqWT were treated with 1 mL of either water or 10 µM CT for 1 h. Total RNA was extracted and purified following the manufacturer’s protocol, and was sent to Novogene Co. for sequencing with an Illumina NovaSeq instrument (poly-A enrichment; PE150). The raw reads were aligned to the *Arabidopsis* TAIR10 reference genome using STAR v2.7.10a [29].The aligned bam files were analyzed with featureCounts v2.0.1 [30], and the count tables for all samples were analyzed with DESeq2 v1.34.0 [31]. GO enrichment and KEGG pathway analyses were performed on PANTHER [32] and KEGG PATHWAY [33] databases, respectively. Significantly regulated genes were defined with the criteria: |log2 fold change| ≥ 1.33 and adjusted *p*-value < 0.05. The adjusted *p*-value was calculated by the DESeq2 package using the built-in Benjamini and Hochberg method with the default FDR cutoff value set as 0.1.

### 2.5. Phosphoproteomic Analysis

#### 2.5.1. Sample Collection

Three hundred roots of SWT or HO from the *cork1-2* mutant line were collected at 0 min, or after treatment with either water or 10 µM CT for 5 or 15 min. Samples were immediately frozen in liquid nitrogen until further analysis.

#### 2.5.2. In-Solution Digest

Tissues were disrupted by using mortar and pestle with liquid nitrogen. Debris were homogenized in lysis buffer (1% (*w*/*v*) SDS, 150 mM NaCl, 100 mM TEAB (triethyl ammonium bicarbonate), and one tablet each of cOmplete Ultra Protease Inhibitor Cocktail and PhosSTOP). After addition of 0.5 µL Benzonase nuclease (250 U/μL), the samples were incubated at 37 °C in a water bath sonicator for 30 min. Proteins were separated from unsolubilized debris by centrifugation (15 min, 18,000× *g*). Each 1.5 mg of total protein per sample was diluted with 100 mM TEAB to gain a final volume of 1.5 mL. Subsequently, cysteine thiols were reduced and carbamidomethylated in one step for 30 min at 70 °C by addition of 30 µL of 500 mM TCEP (tris(2-carboxyethyl)phosphine) and 30 µL of 625 mM 2-Chloroacetamide (CAA). The samples were further cleaned by methanol–chloroform–water precipitation using the protocol of Wessel and Flügge [34]. Protein precipitates were resolubilized in 5% trifluoroethanol of aqueous 100 mM TEAB and digested overnight (18 h) with a Trypsin+LysC mixture (Promega) at a protein-to-protease ratio of 25:1. Each sample was divided in 3 × 0.5 mg used for phosphopeptide enrichment and 150 µg initial protein used for reference proteome analysis. Samples were evaporated in a SpeedVac. The reference proteome sample was resolubilized in 30 µL of 0.05% TFA in H_2_O/acetonitrile (ACN) 98/2 (*v*/*v*) filtered through 10 kDa MWCO PES membrane spin filters (VWR). The filtrate was transferred to HPLC vials and injected into the LC-MS/MS instrument.

#### 2.5.3. Phosphopeptide Enrichment

Phosphopeptides were enriched by using TiO_2_+ZrO_2_ TopTips (Glygen Corp., Columbia, MD, USA). TopTips were loaded with 0.5 mg protein isolate using 3 TopTips per biological replicate after equilibration with 200 µL Load and Wash Solution 1, LWS1 (1% trifluoroacetic acid (TFA), 20% lactic acid, 25% ACN and 54% H_2_O). TopTips were centrifuged at 1500 rpm (~200× *g*) for 5 min at room temperature. After washing with 200 µL LWS1, the TiO_2_/ZrO_2_ resin was washed with 25% ACN, and, subsequently, the phosphopeptides were eluted with 200 µL NH_3_·H_2_O (NH_4_OH), pH 12. The alkaline solution was immediately evaporated using a SpeedVac. The phosphoproteome samples were resolubilized in 50 µL of 0.05% TFA in H_2_O/ACN 98/2 (*v*/*v*) filtered through 10 kDa MWCO PES membrane spin filters (VWR). The filtrate was also transferred to HPLC vials and injected into the LC-MS/MS instrument.

#### 2.5.4. LC-MS/MS Analysis

Each sample was measured in duplicate (2 analytical replicates of 3 biological replicates of a reference proteome fraction and a phosphoproteome fraction). LC-MS/MS analysis was performed on an Ultimate 3000 nano RSLC system connected to a QExactive HF mass spectrometer (both Thermo Fisher Scientific, Waltham, MA, USA). Peptide trapping for 5 min on an Acclaim Pep Map 100 column (2 cm × 75 µm, 3 µm; Thermo Fisher Scientific) at 5 µL/min was followed by separation on an analytical Acclaim Pep Map RSLC nano column (50 cm × 75 µm, 2 µm; Thermo Fisher Scientific). Mobile-phase gradient elution of eluent A (0.1% (*v*/*v*) formic acid in water) mixed with eluent B (0.1% (*v*/*v*) formic acid in 90/10 ACN/water) was performed using the following gradient: 0–5 min at 4% B, 30 min at 7% B, 60 min at 10% B, 100 min at 15% B, 140 min at 25% B, 180 min at 45% B, 200 min at 65% B, 210–215 min at 96% B and 215.1–240 min at 4% B. Positively charged ions were generated at a spray voltage of 2.2 kV using a stainless steel emitter attached to a Nanospray Flex Ion Source (Thermo Fisher Scientific). The quadrupole/orbitrap instrument was operated in Full MS/data-dependent MS2 Top15 mode. Precursor ions were monitored at *m*/*z* 300–1500 at a resolution of 120,000 FWHM (full width at half maximum) using a maximum injection time (ITmax) of 120 ms and an AGC (automatic gain control) target of 3 × 10^6^. Precursor ions with a charge state of z = 2–5 were filtered at an isolation width of *m/z* 1.6 amu for further HCD fragmentation at 27% normalized collision energy (NCE). MS2 ions were scanned at 15,000 FWHM (ITmax = 100 ms, AGC = 2 × 10^5^) using a fixed first mass of *m/z* 120 amu. Dynamic exclusion of precursor ions was set to 30 s. The LC-MS/MS instrument was controlled by Chromeleon 7.2, QExactive HF Tune 2.8 and Xcalibur 4.0 software.

#### 2.5.5. Protein Database Search

Tandem mass spectra were searched against the UniProt database (https://www.uniprot.org/proteomes/UP000006548, accessed on 6 January 2022) of *A. thaliana* using Proteome Discoverer (PD) 2.4 (Thermo Fisher Scientific) and the Sequest HT algorithm. Two missed cleavages were allowed for tryptic digestion. The precursor mass tolerance was set to 10 ppm, and the fragment mass tolerance was set to 0.02 Da. Modifications were defined as dynamic Met oxidation, phosphorylation of Ser, Thr and Tyr, protein N-term acetylation with and without Met-loss as well as static Cys carbamidomethylation. A strict false discovery rate (FDR) ≤ 1% (peptide and protein level) and an X_corr_ score ≥ 4 were required for positive protein hits. The Percolator node of PD2.4 and a reverse decoy database were used for q-value validation of spectral matches. Only rank 1 proteins and peptides of the top-scored proteins were counted. Label-free protein quantification was based on the Minora algorithm of PD2.4 using precursor abundance based on intensity and a signal-to-noise ratio ≥ 5. Normalization was performed by using the total peptide amount. Imputation of missing quantification (quan) values was applied by using abundance values of 75% of the lowest abundance identified per sample. For the reference proteome analysis used for master protein abundance correction of the phosphoproteome data, phosphopeptides were excluded from quantification. Differential protein and phosphopeptide abundance was defined as a fold change of ≥2, ratio-adjusted *p*-value ≤ 0.05 (*p*-value/log4ratio) and identified in at least 2 of 3 replicates of the sample group with the highest abundance. Division by the log4ratio ensures that the adjusted *p*-value increases whenever the fold change is ≤4. This adjustment requires moderate fold changes (2–4) to have a stricter statistical significance level. On the other hand, the adjusted *p*-value decreases when the fold change is ≥4 in order to limit the possibility that the data quality (e.g., due to technical variation) is overrated when estimating the significance of a difference between replicate values of two comparison groups.

### 2.6. ROS and [Ca^2+^]_cyt_ Measurements

Seedlings were grown vertically on Hoagland agar medium (Hoagland’s No. 2 Basal Salt Mixture; Sigma-Aldrich) for 16 days before harvesting the leaf discs (about 1 mm in diameter) or approximately 70% of the roots for ROS and [Ca^2+^]_cyt_ measurements [35,36].

For ROS measurement, root tissue was incubated in sterile water in a 96-well plate in the dark at room temperature for 1 h. Prior to elicitor treatment, water was replaced by 150 μL of assay solution containing 2 μg/mL horseradish peroxidase (Sigma-Aldrich) and 100 μM luminol (FUJIFILM Wako Chemicals Europe GmbH, Neuss, Germany).

The [Ca^2+^]_cyt_ concentration was inferred from aequorin-based luminescence [24]. Leaf discs and root tissue were incubated overnight in 150 μL of 7.5 μM coelenterazine solution (P.J.K. GmbH, Kleinblittersdorf, Germany) in a 96-well plate in the dark at room temperature.

Bioluminescence counts from elicitor application were recorded as relative light units (RLU) with microplate luminometer (Luminoskan Ascent v2.4, Thermo Fisher Scientific, or Mithras LB940, Berthold Technologies, Bad Wildbad, Germany).

Cellobiose (C7252, Sigma-Aldrich, Germany), cellotriose (C1167, Sigma-Aldrich, or 0-CTR-50MG, Megazyme, Wicklow, Ireland) and chitohexaose (OH07433, Carbosynth, Berkshire, United Kingdom) were used as elicitors. Concentration of elicitors, unless specified, was 10 μM for cellotriose and chitohexaose and 1 mM for cellobiose. All elicitors were dissolved and diluted with distilled water.

### 2.7. Nucleic Acid Isolation, PCR and RT-qPCR

Plant tissue was homogenized in liquid nitrogen. DNA extraction was performed according to Doyle [37]. RNA extraction was done with Trizol^TM^ reagent (Thermo-Fisher Scientific), treated with Turbo DNA-free^TM^ Kit (Thermo-Fisher Scientific), and reverse transcribed with RevertAid Reverse Transcriptase (Thermo-Fisher Scientific) according to the manufacturer’s instructions.

Genotyping of the back-crossed F2 mutant population was achieved by PCRs with genomic DNA. PCRs were run with DreamTaq DNA Polymerase (Thermo Fisher Scientific) in a thermal cycler (Applied Biosystems SimpliAmp Thermal Cycler, Thermo Fischer Scientific). Quantitative reverse transcription PCRs (RT-qPCRs) were performed with Dream Taq DNA Polymerase (Thermo-Fisher Scientific, Germany) with the addition of Evagreen^®^ (Biotium, Fremont, CA, USA). CFX Connect^TM^ Real-Time PCR Detection System (Bio-Rad, Feldkirchen, Germany) was used for running and analyzing qPCRs. The expression of genes was normalized to a housekeeping gene encoding a ribosomal protein (RPS; AT1G34030). The resulting ΔCq values were used for statistical analysis. For the confirmation of SNP in the EMS mutant, a primer pair flanking the SNP site was designed, and the region was amplified with Phusion^TM^ High-fidelity DNA polymerase (Thermo-Fisher Scientific). The PCR product was purified with NucleoSpin Gel and PCR Clean-up kit (Macherey-Nagel, Düren, Germany) and sequenced by Eurofins Genomics, Ebersberg, Germany. All primers used are listed in Appendix A.

### 2.8. Multiple Sequence Alignment

Amino acid sequences of malectin RLKs and malectin-like RLKs were retrieved from the Uniprot database and aligned with MEGA7 [38] using the default Clustal W algorithm [39]. The aligned sequences were edited for presentation using BioEdit v7.2.5 (http://www.mbio.ncsu.edu/BioEdit/bioedit.html). The accession numbers for all sequences are listed in Appendix A.

### 2.9. Plasmid Construction

Full-length coding regions of *ARF1* and *CORK1* (*AT1G56145.1*) were amplified from the reverse-transcribed RNA (cDNA) using Phusion^TM^ high-fidelity DNA polymerase (Thermo-Fisher Scientific). The fragments were cloned into entry vector pENTR^TM^/D-TOPO^TM^ and transferred to pB7FWG2.0 destination vector [40] with Gateway™ LR Clonase™ II (Thermo-Fisher Scientific). Site-directed mutagenesis was carried out to specifically mutate the amino acid residues of interest. For complementation experiments, two stop codons were added at the end of the gene of interest. The two stop codons were removed to generate a CORK1–GFP fusion protein with or without point mutations in the malectin domain.

For the kinase activity assay, the cytoplasmic domain of CORK1 (residues 654-1039; CORK1^KD^) was cloned and ligated into the expression vector pET28a using restriction enzymes *Bam*HI and *Eco*RI. Two stop codons were added before the *Eco*RI restriction site, generating a 6X His-tagged protein at the N-terminus. The mutated form (CORK1^KD-G748E^) was obtained by site-directed mutagenesis. The mutated PCR fragment for the kinase domain was cloned and ligated into the expression vector pGEX1λT using restriction enzymes *Bam*HI and *Eco*RI, generating a glutathione S-transferase (GST) fusion protein at the N-terminus.

To generate the luciferase reporter constructs with the *WRKY30* and *WRKY40* promoters, 2 kb DNA fragments upstream of the respective start codons were cloned and ligated into the pJS plasmid [41] using *Nco*I and *Bam*HI restriction sites.

For every construct, the insert sequence was confirmed by Sanger sequencing (Eurofins Genomics). Primers used are listed in Appendix A.

### 2.10. Protein Expression, Extraction, Purification and Kinase Assay

The pET28a vector with CORK1^KD^ and the pGEX1λT vector with CORK1^KD-G748E^ were transformed into *E. coli* strain BL21(DE3) pLysS (Novagen, Darmstadt, Germany). For the expression of CORK1^KD^, the transformed bacteria were grown directly in LB broth [42] with 34 μg/mL chloramphenicol and 50 μg/mL kanamycin at 37 °C for 16 h with shaking. IPTG (isopropyl β-D-1-thiogalactopyranoside; Carl Roth, Karlsruhe, Germany) was added to the culture to a final concentration of 1 mM to induce protein expression for 3 h. For the expression of GST-CORK1^KD-G748E^, the overnight culture was inoculated into LB broth with 34 μg/mL chloramphenicol and 100 μg/mL ampicillin at 37 °C with shaking. After O.D._600nm_ reached 0.6, IPTG was added to the broth to a final concentration of 1 mM to induce protein expression for 3 h at 25 °C. Cells were collected by centrifugation for 10 min at 4 °C, 4193× *g*.

The bacterial pellet for CORK1^KD^ was resuspended in extraction buffer containing 50 mM Tris-HCl, pH 8.0, 300 mM NaCl and 0.1% (*w*/*v*) CHAPS (3-[(3-cholamidopropyl)-dimethylammonio]-1-propanesulfonate hydrate, Sigma-Aldrich). Sonication was applied to lyse the cells. Cell debris was pelleted by centrifugation for 10 min at 4 °C, 12,000 rpm. Cell lysate was incubated with ProBond Ni-NTA resin (Thermo-Fisher Scientific) for 0.5–1 h. The resin was washed in the same buffer with 20 mM imidazole 3 times to remove unbound protein. Finally, His-tagged protein was eluted with the same buffer containing 250 mM imidazole. For GST-CORK1^KD-G748E^, purification was done with a Pierce™ GST Spin Purification Kit following the manufacturer’s instructions. Purified proteins were concentrated and buffer-exchanged in kinase assay buffer (25 mM Tris-HCl, pH 7.5, 10 mM MgCl_2_) using a Vivaspin^®^ 20 ultrafiltration unit (3000 MWCO, Sartorius, Göttingen, Germany).

The kinase activity assay was carried out by mixing 2 μg of CORK1^KD^ or GST-CORK1^KD-G748E^ with 3 μg of myelin basic protein (MBP) (Sigma-Aldrich, Germany) in kinase reaction buffer (25 mM Tris-HCl, pH 7.5, 10 mM MgCl_2_, 1 mM DTT and 100 μM ATP). The reaction mixtures were incubated at 30 °C for 30 min and terminated by adding SDS-PAGE loading buffer. The proteins were separated by SDS-PAGE. Protein phosphorylation was examined by staining with Pro-Q Diamond phosphoprotein gel stain (Thermo-Fisher Scientific, Germany) following the manufacturer’s instructions, or with morin hydrate [43], and visualized using AlphaImager HP system (ProteinSimple, San Jose, CA, USA). Coomassie blue staining (Roti-Blue, Carl Roth) was conducted to visualize the total protein.

### 2.11. Complementation of the COM Receptor Mutant

*CORK1* or *ARF1* in pB7FWG2.0 vector were transformed into the COM non-responsive EMS mutant EMS71 using the floral dip method with *Agrobacterium tumefaciens* strain GV3101 [44]. Complemented plants were selected on soil using a 0.1% (*v*/*v*) BASTA solution 14 and 18 days after sowing.

### 2.12. Transient Expression in A. thaliana

Transient co-expression of the pFRK1::luciferase reporter with the receptor expression constructs in mesophyll protoplasts of *A. thaliana* Col-0 wild-type was performed as described previously [41,45]). Luminescence was recorded for up to 5 h in W5 medium containing 200 µM firefly luciferin (Synchem UG, Felsberg, Germany) after overnight incubation for 14 h and subsequent treatment with CT or control solutions. After the measurements, protoplasts were harvested by centrifugation and denatured in 2 × SDS-PAGE loading buffer. The crude extracts were separated on 8% polyacrylamide gels (acrylamide: bis-acrylamide ratio of 29:1) and transferred to nitrocellulose membranes. Membranes were saturated with 5% milk powder in PBS with 0.05% Tween-20 (PBS-T) followed by immunostaining with anti-GFP antibodies (1:5000 in PBS-T; Torrey Pines Biolabs, Secaucus, NJ, USA) and secondary goat–anti-rabbit antibodies coupled to alkaline phosphatase (Applied Biosystems, Thermo Fisher Scientific) using CDP-star as substrate.

### 2.13. Microscopy

Protoplasts of *A. thaliana* were mounted on a glass slide with cover slip for microscopic inspection using an Axio Imager.M2 (Zeiss Microscopy GmbH, Jena, Germany). Bright-field and fluorescent images were recorded with a monochromatic camera: Axiocam 503 mono (Zeiss Microscopy GmbH). Confocal microscopy was performed according to Tseng et al. [46]. The plant plasma membrane was stained with the dye RH414 (N-(3-Triethylammoniumpropyl)-4-(4-(4-(Diethylamino)phenyl)Butadienyl)Pyridinium Dibromide; Thermo Fischer Scientific). Digital images were processed with the ZEN software (Zeiss Microscopy GmbH).

### 2.14. Statistical Tests

Statistical tests were performed using R Studio v1.1.463 with R v4.1.2. Figures were plotted using Python 3.7.4 and arranged with LibreOffice Draw 5.1.6.2.

### 2.15. Data Availability

Raw sequences for the GWAS have been deposited to the NCBI Gene Expression Omnibus (GEO) database (accession no. GSE197891). For transcriptome analysis, raw sequences and the count tables after DESeq2 analysis have been deposited to the Gene Expression Omnibus (GEO) database (accession no. GSE198092). Lists of differentially expressed genes mentioned here are provided in Appendix A. The mass spectrometry proteomics data have been deposited to the ProteomeXchange Consortium via the PRIDE partner repository [47] with dataset identifier PXD033224. Lists of significantly changed phosphopeptides mentioned here are provided in Appendix A.

## 3. Results

### 3.1. Identification of CelloOligomer Receptor Kinase 1 (CORK1)

To identify proteins involved in COM perception, an EMS-treated seedling population generated from the wild-type pMAQ2 aequorin line (AeqWT) was screened. Roots from individual F2 seedlings were used to monitor [Ca^2+^]_cyt_ elevation upon 1 mM cellobiose (CB) application. One mutant (designated as EMS71) showed no [Ca^2+^]_cyt_ elevation in response to CB (1 mM) and CT (10 μM; data not shown). The non-responsive phenotype was confirmed in the F3 generation in both root and leaf tissues (Figure 1A,B). Since [Ca^2+^]_cyt_ elevation induced by chitin was not affected (Figure 1C), EMS71 is specifically impaired in COM perception.

EMS71 was back-crossed to AeqWT, and the F2 population was divided into responsive and non-responsive groups. DNA from these two groups was extracted, sequenced and analyzed as described in the Materials and Methods. Finally, two candidate genes, *AT1G59750* (*ARF1*, A248V) and *AT1G56145* (*CORK1*, G748E), were selected for further confirmation.

The two candidate genes were overexpressed with CaMV 35S promoter in EMS71. Upon CT application, [Ca^2+^]_cyt_ elevation was only detected in EMS71 seedlings transformed with the *CORK1* construct, but not in those with *ARF1*, suggesting *CORK1* is required for the [Ca^2+^]_cyt_ elevation induced by COMs (Figure 1D).

In addition, two T-DNA insertion mutant lines, *cork1-1* and *cork1-2*, were crossed to AeqWT for [Ca^2+^]_cyt_ measurements. In the F2 generation, ~25% of the seedlings were non-responsive to CT (Figure 2A). Genotyping showed that the responsive seedlings were either segregated wild-type (SWT) or heterozygotes, while non-responsive seedlings were all homozygous (HO) for the T-DNA insertion (Figure 2B). The *CORK1* transcript level was significantly reduced in the HO seedlings compared to SWT seedlings (Figure 2C). This demonstrates that CORK1 mediates COM perception in *Arabidopsis*.

The gene model for *CORK1* predicts three RNA isoforms (Figure 2D). AT1G56145.1 (lacking an intron near the 3′ end), AT1G56145.2 (deduced from the complete DNA sequence) and AT1G56145.3 (omitting the first 309 nucleotides from the 5′ end). In the *cork1-1* and *cork1-2* mutants, the T-DNAs were inserted in the first exon and the exon located near the 3′ end, respectively (Figure 2D). The SNP for EMS71 is caused by a G→A exchange of the 2243^rd^ nucleotide, converting the 748th glycine residue to glutamic acid (Figure 2D,E). Thus, the single mutation affects all three predicted RNA isoforms.

Based on the sequence of the first isoform, CORK1 is annotated as a leucine-rich repeat transmembrane protein kinase with a predicted 24 amino acid long signal peptide at the N-terminus, followed by leucine-rich repeat (LRR) domains and a malectin domain (MD). After the transmembrane domain, a Ser-Thr/Tyr kinase domain is predicted to reside in the cytoplasm (Figure 2E).

### 3.2. CORK1 Encodes a Functional Receptor Kinase

To determine whether *CORK1* encodes a functional LRR receptor kinase, subcellular localization was first examined by transfecting *A. thaliana* protoplasts with a 35S::CORK1-GFP construct. The GFP signal at the plasma membrane confirmed that CORK1 is a membrane-associated protein (Figure 3A). In the stable transformed EMS71 mutant lines overexpressing CORK1–GFP fusion protein, the GFP signal is also located at the plasma membrane, confirming the result from the transient assay (Figure 6C).

Next, the cytoplasmic region encompassing the kinase domain (CORK1^KD^) was expressed with a N-terminus polyhistidine tag (6x-His) to characterize its kinase activity. Similarly, the point mutation was introduced (CORK1^KD-G748E^) and expressed with a N-terminus Glutathione-S-Transferase (GST) tag to test whether the mutation found in the kinase domain of EMS71 affects the kinase activity. Figure 3B shows that the substrate myelin protein bovine (MBP) was only phosphorylated by 6x-His-CORK1^KD^ but not by the mutated form GST-CORK1^KD-G748E^. At the same time, CORK1^KD^ exhibited strong autophosphorylation. This suggests that *CORK1* encodes a functional kinase domain, and the G748E mutation could have disrupted the kinase activity.

### 3.3. cork1 Mutant Failed to Produce ROS upon COM Perception

Besides [Ca^2+^]_cyt_ elevation, COMs also induce ROS production, albeit less than classical PAMPs such as chitin [13]. In SWT roots, but not those in HO, of *cork1-1* and *cork1-2* seedlings, ROS was produced after CT treatment. ROS production was detected upon chitin treatment in both SWT and HO (Figure 4A,B). This indicates that CORK1 is required for COM- but not chitin-induced ROS production.

### 3.4. Upregulation of WRKY30 and WRKY40 mRNA Levels by COMs Is CORK1-Dependent

Since CB activates *WRKY30* and *WRKY40* expression [11,13], we checked whether the activation of these genes requires *CORK1*. CT and chitin were applied to roots of SWT and HO seedlings of *cork1-1* and *cork1-2*. After 1 h, the *WRKY30* and *WRKY40* transcript levels in SWT of both T-DNA lines were upregulated ~30- and ~15-fold, respectively (Figure 5A,B). On the other hand, no significant response to CT was observed in the HO mutants (Figure 5A,B). Chitin stimulated the *WRKY30* transcript level ~10-fold and that of the WRKY40 ~2-fold in both genotypes (Figure 5A,B). This demonstrates that COM-mediated activation of *WRKY30* and *WRKY40* requires CORK1.

### 3.5. Two Phe Residues in the Malectin Domain Are Important for CT Response

Sequence alignment of the *A. thaliana* LRR-MD RLKs demonstrated that two Phe residues within the MD (F520 and F539) are highly conserved in all MD RLKs (Figure 6A and Appendix A) and malectin-like (MLD) RLKs (Appendix A). It has been suggested that aromatic rings of amino acids interact with the apolar side of carbohydrates [48,49,50]. Therefore, we changed the two conserved Phe residues to Ala. In the EMS71 mutant transformed with a 35S::CORK1-GFP construct, [Ca^2+^]_cyt_ elevation in response to CT application was restored. The [Ca^2+^]_cyt_ response in plants transformed with either of the two Phe mutant versions (F520A or F539A) was significantly reduced, and nearly no [Ca^2+^]_cyt_ could be observed in plants transformed with the double-mutated version (Figure 6B). However, the lack of calcium elevation was not due to the localization change or the absence of the CORK1–GFP fusion protein, as GFP signal was present at the plasma membrane (Figure 6C). To further support the importance of the two Phe residues, mesophyll protoplasts of *A. thaliana* were co-transformed with the pFRK1::luciferase reporter and either the wild-type or the double-mutated version of CORK1. The co-expression of wild-type CORK1 with the reporter gene conferred responsiveness to treatment with 1 µM CT, which was absent when the mutated form was expressed (Figure 6D–G). This suggests that the two conserved Phe residues are important in COM-induced responses in *A. thaliana.*

**Figure 6 cells-11-02960-f006:**
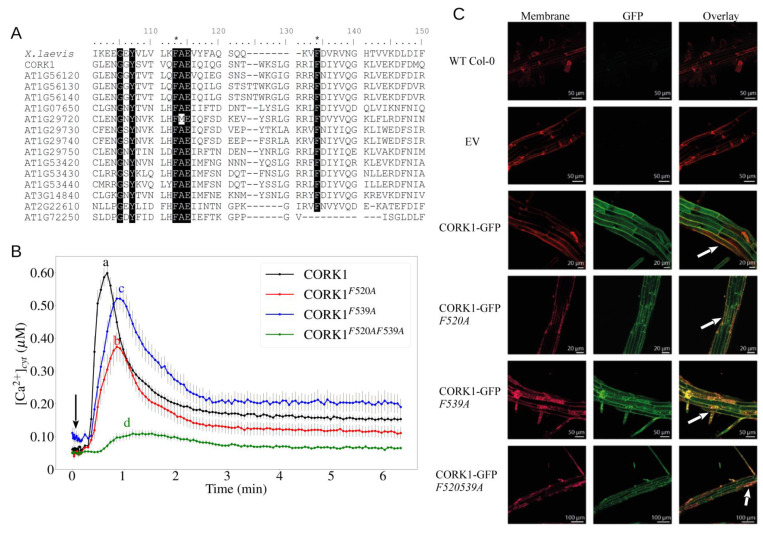
Two conserved phenylalanine residues in the malectin domain of CORK1 are important for COM perception. (**A**) Alignment of the amino acid sequences from malectin of *X. laevis* and the malectin domains present in *A. thaliana* LRR-malectin RLKs. Shown here are amino acids from position 101–150 of the alignment. Black shade indicates conserved amino acid residues over 90% threshold. The two conserved phenylalanine residues are indicated with an asterisk. (**B**) Cytoplasmic calcium elevation by 10 µM CT in root tissue of EMS71 complemented with CORK1, or with single (CORK1^F520A^/CORK1^F539A^) or double (CORK1^F520AF539A^) mutation in the two conserved phenylalanine residues. Error bars represent SE from 8 seedlings. Arrow indicates the onset of elicitor application. Statistical significance at the peak value was determined by Tukey’s HSD test with *p* ≤ 0.05 and is indicated by different lowercase letters. The experiment was repeated 3 times with similar results. (**C**) GFP signal in the roots of the wild-type Col-0 (WT) and the transformed EMS71 mutants. EV: Empty vector; EMS71 transformed with the construct 35S::CORK1 with two stop codons after the coding region. The plasma membrane was stained with RH414, and the overlapping signals from GFP and RH414 at the plasma membrane are indicated with white arrows. (**D**–**G**) Protoplasts from *A. thaliana* Col-0 were transfected with the (**D**) pFRK1::Luciferase (pFRK1::LUC) reporter construct, (**E**) the reporter construct plus the CORK1 receptor or (**F**) the reporter construct plus the double-mutated version CORK1^F520AF539A^. Results show luciferin-dependent light emission over time after treatment with water (Mock) or 1 µM CT. Arrow indicates the onset of elicitor application at 0 h. Each datapoint represents the mean value from 4 technical replicates. Error bars represent SE. Statistical significance was determined by Student’s T-test between the two treatments (* *p* ≤ 0.05; ** *p* ≤ 0.01). The experiment was repeated 4 times with similar results. (**G**) Western blot indicating the expression of both versions of GFP-tagged CORK1; ctrl: control, protoplast transfected with pFRK1::LUC reporter construct only.

### 3.6. Transcriptome Analysis Uncovered COM/CORK1 Responsive Genes

To identify the biological functions of COMs, we performed transcriptome analysis with the roots of *cork1-2* SWT and HO seedlings 1 h after the application of either 10 μM CT or water. Among the 23106 mapped genes, 561 genes were up- and 54 genes downregulated by CT in a *CORK1*-dependent manner. On the contrary, only 2 genes were significantly upregulated and no genes were downregulated by CT in HO (Figure 7A; Appendix A). This shows that the responses to COMs are highly specific to *CORK1*.

Gene ontology (GO) enrichment analysis showed a profound increase in genes involved in tryptophan biosynthesis, cell wall modification and secondary metabolite production (Appendix A). The genes encoding ASA1 (anthranilate synthase α subunit 1; *AT5G05730*) and ASB1 (anthranilate synthase β subunit 1; *AT2G25220*), which carry out the first step in the tryptophan biosynthesis from chorismate, were ~10-fold upregulated, and genes encoding TSA1 (tryptophan synthase α chain; *AT3G54640*), which catalyzes the last step in the biosynthesis, were ~8-fold upregulated by CT (Table 1 and Figure 7B).

Among the first 15 categories for the most strongly regulated genes, 5 categories are related to “cell wall” functions. All of them center around callose deposition and cell wall thickening, and most of these genes/proteins are described in the context of defense (Appendix A). Genes in these categories encode *FLS2* (~4 fold; *AT5G46330*), *MYB51* (~13 fold; *AT1G18570*), *UDP-glycosyltransferase 74B1* (~4.5 fold; *AT1G24100*), the cytochrome P450 enzymes *CYP81F2* (~12 fold; *AT5G57220*) and *CYP83B1* (~7 fold; *AT4G31500*) as well as the ABC-transporter gene *ABCG36* (~3 fold; *AT1G59870*). Similarly, genes involved in lignin biosynthesis (phenylpropanoid metabolism) were also upregulated, such as those for cinnamate-4-hydroxylase (~3 fold; C4H, *AT2G30490*), 4-coumarate-CoA ligase 1 (~4 fold; 4CL1, *AT1G51680*), phenylalanine ammonia lyase 1 (~3 fold; PAL1, *AT2G37040*), and for three enzymes important for lignin production: caffeoyl-CoA 3-O-methyltransferase (~13 fold; CCoAOMT, *AT1G67980*), cinnamyl alcohol dehydrogenase 5 (~3 fold; CAD5, *AT4G34230*) and peroxidase 4 (~5 fold; class III peroxidase PER4, *AT1G14540)* [51,52]. *PEN2* and *PMR4/GSL5,* encoding a myrosinase and a callose synthase, respectively, were only slightly upregulated (~1.7 fold; Table 1 and Figure 7B).

In addition to cell-wall-related genes, *SOT16* and *SOT17*, which encode sulfotransferases for glucosinolate production, were upregulated ~5.5 fold. Likewise, the transcript level for *CYP71B15* (*PAD3*; *AT3G26830*), which is required for camalexin production, was ~3.5-fold upregulated. In line with qPCR analyses and previous reports [11,13], *WRKY30, WRKY40* and the lipoxygenase genes involved in jasmonic acid synthesis, *LOX1* (~3.5 fold; *AT1G55020*), *LOX3* (~11.5 fold; *AT1G17420*) and *LOX4* (~5.5 fold; *AT1G72520*), responded to COMs. Finally, genes involve in the FLS2 signaling pathway were upregulated, such as *FRK1* (~4.5 fold; *AT2G19190*) and *WRKY29* (~2.5 fold; *AT4G23550*) (Table 1 and Figure 7B) [53].

Interestingly, most of the downregulated genes are involved in ion homeostasis or defense, such as a *PR-1-like* gene (*AT2G19990*, ~0.16 fold) (Figure 7B and Appendix A).

In summary, COM induces genes possibly involved in cell wall reinforcement, defense-related secondary metabolite synthesis, and crosstalk with other signaling components, and these responses require CORK1.

### 3.7. COM/CORK1-Mediated Changes in the Phosphoproteome Pattern in Roots

To identify early COM/CORK1 targets, the phosphoproteomes of SWT and HO roots were analyzed 5 and 15 min after 10 μM CT or water (control) application. Most of the proteins with a significantly altered phosphorylation state are related to (i) the function of cellulose synthase complex (CSC) and its translocation to the plasma membrane, (ii) the ER secretory pathway and protein sorting, (iii) signal transduction, or (iv) defense/stress responses (Appendix A).

Cellulose synthases 1 and 3 (CESA1, -3) of the CSC are required for cellulose synthesis for the primary cell wall, and the protein is rapidly phosphorylated at Ser24 and Ser176, respectively, in response to CT application. Mutations of CESA phosphorylation sites modulate anisotropic cell expansion and bidirectional mobility of the cellulose synthase [54]. Besides CESAs, we also identified cellulose synthase-interactive 1 (CSI1) as a phosphorylation target of CT at Thr37. Association of CSC with cortical microtubules is mediated by CSI1, and the protein contains multiple phosphorylation sites potentially involved in regulatory processes [55]. Loss-of-function CSI1 mutants are impaired in the dissociation of the CSC from the microtubules during their passage to the plasma membrane, which results in cellulose deficiency in the mutant cell walls [56]. COMPANION OF CELLULOSE SYNTHASE 1 and 2 (CC1/CC2) and the N-terminal domain in CSI are responsible for the connection of the CSCs to the cortical microtubules, and *csi1* mutants have impaired microtubule stability under salt stress [57,58], but also for CSC delivery to the plasma membrane and its recycling [59,60]. In addition to its role in trafficking and mobility of CSCs, microtubules also influence the orientation and crystallinity of cellulose [60]. Thus, CESA1 and CSI1 are two central players in the cellulose repair mechanism, and their phosphorylation by COM/CORK1 signaling could alter cell wall biosynthesis.

Among the proteins involved in the endomembrane system and the secretory pathway are two GTPases that regulate membrane trafficking: AGD5, a GTPase-activating protein operating at the *trans*-Golgi network [61], and RABA5C, a GTPase that specifies a membrane trafficking pathway to geometric edges of lateral root cells [62]. The 1-phosphatidylinositol-3-phosphate 5-kinase is involved in maintenance of endomembrane homeostasis, including endocytosis and vacuole formation [63]; the exocyst complex component SEC8 participates in the docking of exocytic vesicles with fusion sites on the plasma membrane and the formation of new primary cell wall; the vacuolar sorting protein 41 regulates vacuolar vesicle fusions and protein sorting together with phosphoinositides [64]; a SNARE protein as part of a complex facilitates trafficking in the endomembrane system, including distinct secretory and vacuolar trafficking steps. The phosphorylated myosin mediates the organization of actin filament and vesicle transport along the filaments. Finally, MAPK 17 influences the number and cellular distribution of peroxisomes through the cytoskeleton–peroxisome connection [65]. Not surprisingly, stimulation of membrane trafficking, protein sorting and secretion also affects enzymes involved in biosynthesis of cellulose, callose and other polysaccharides, such as CESAs, CSI1, callose synthase and a regulator of callose deposition, and the PAMP-induced coiled coil protein AT2G32240 [66]. The cytosolic UDP-glucuronic acid decarboxylases, AT3G46440 and AT5G59290, produce UDP-xylose, which is a substrate for many cell wall carbohydrates, including hemicellulose and pectin. UDP-xylose is also known to feedback regulate several cell wall biosynthetic enzymes, many of them associated with the endomembrane system [67,68,69]. Phosphorylated proteins with related functions control auxin translocation at the plasma membrane (AT1G56220, ABCG36).

Inspection of target proteins at different time points uncovered that the canonical immunity-related mitogen-activated protein kinases MPK3 and MPK6 showed increased phosphorylation (Ser16 of MPK3 and Tyr223 of MPK6) at both time points after CT application in SWT roots. Likewise, the calmodulin-binding protein IQM4 (at Ser505, Ser509, Ser520 and Ser525) and Ca^2+^-dependent protein kinase CPK9 (at Ser69) were among the most phosphorylated targets at both time points and link stress-induced Ca^2+^ signaling to abscisic acid [70,71]. On the other hand, phosphorylation of the plasma membrane-localized FERONIA (at Ser695), SERK1 (at Thr450 and Thr463) and MAPKKK3, which mediates MPK3/6 activation by at least four pattern-recognition receptors (FLS2, EFR, CERK1 and PEPRs) [72], was only detectable at the early time point. Phosphorylation at Ser716 of tumor necrosis factor receptor-associated factor (TRAF) 1B, which is involved in immune receptor turnover, was increased 5 min after CT treatment, but was decreased 15 min after CT treatment (Table 2) [73].

Among the defense-related proteins, phosphorylation of RBOHD at Ser347 was significantly decreased at the early time point (Table 2). The MPK8 module negatively regulates ROS accumulation through controlling expression of the *RBOHD* gene, and the phosphorylation state decreased 15 min after CT application. Takahashi et al. [74] proposed that Ca²^+^/CaMs and the MAP kinase phosphorylation cascade converge at MPK8 to monitor or maintain ROS homeostasis. Likewise, phosphorylation of JOX2 at Ser369, which catalyzes the hydroxylation of jasmonic acid to 12-OH-jasmonic acid and thus restricts the generation of the active jasmonic acid-isoleucine, was only detectable at the early time point. Further, EXA1 (essential for potexvirus accumulation 1), controlling virus infection, was only phosphorylated at Ser1553 at the early time point [75]. In contrast, phosphorylation of EDR4 (enhanced disease resistance 4) which represses salicylic-acid-mediated resistance, and ZAT10, a repressor of abiotic stress and jasmonic acid responses, was only stimulated 15 min after CT application [76,77]. The ABC transporter G36, which controls pathogen entry into cells, was phosphorylated at both time points, and its mRNA was upregulated in our expression analysis (Table 1). The identified targets of COM/CORK1 signaling show dynamic changes in the protein phosphorylation pattern. Considering the different roles of these proteins in activating or repressing signaling and defense responses, it can be speculated that COM/CORK1 signaling establishes a moderate immune response and maintains its homeostasis.

## 4. Discussion

We demonstrate that CORK1, a leucine-rich repeat-malectin receptor kinase, is required for COM-mediated rapid increase in [Ca^2+^]_cyt_ level and stimulation of ROS production in *Arabidopsis*. Transcriptome analyses uncovered CT-regulated and CORK1-dependent target genes of a proposed COM/CORK1 signaling pathway. Major CT/CORK1 target genes are involved in cell wall strengthening, e.g., by activating genes in callose deposition, secondary metabolite metabolism and Trp biosynthesis. Phosphoproteome analysis identified early COM/CORK1 target proteins involved in secretory pathways and vesicle trafficking; plasma membrane-associated RBOHD, FER and SERK1; cytoplasmic MPK3/6; novel MAPKs such as MAPKKK3, MPK17 and MPK8; as well as downstream proteins involved in plant immunity. Interestingly, the different phosphorylation patterns 5 and 15 min after the stimulus, and phosphorylation of EXA1, TRAF1B, MPK8, JOX2 and EDR4, demonstrate that CT establishes a balanced defense response. For instance, CT stimulates ROS production. Simultaneously, phosphorylation and thus activation of RBOHD was repressed at the early time point (Table 2), and MPK8, which is involved in establishing ROS homeostasis [74], is phosphorylated. Furthermore, expression of defense-related genes such as *WRKY30*/*40* is stimulated by COM, while phosphorylation of EXA1, TRAF1B, JOX2 and EDR4 could restrict or balance defense responses. Crosstalk to other receptor kinases is demonstrated by FER and SERK1, and potentially by the upregulation of *FLS2* at the mRNA level. The observed downstream responses are consistent with the idea that COM/CORK1 activates processes that maintain cell wall integrity.

### 4.1. LRR-Malectin Receptors Are the New Players in Cell Wall Surveillance

Malectins were first discovered in *Xenopus* [78]. They are located in the ER of animal cells and bind to diglucosylated N-linked glycans to control glycoprotein quality [79,80,81]. The ligands of malectins include the disaccharide maltose and nigerose [50,78]. The structure of malectins is similar to the carbohydrate-binding modules found in enzymes that degrade the plant cell wall [78,82,83]. Although the two Phe residues conserved in all MDs in *A. thaliana* are not conserved in the maltose-binding malectin from *Xenopus* [50], the replacement of the two Phe with Ala eliminated CT-induced [Ca^2+^]_cyt_ elevation and reporter gene activation (Figure 6). This suggests a functionally divergent role of malectins in plants and animals. The two Phe residues in the plant MDs might be specifically involved in binding COMs with β 1-4-bonds, while non-plant malectins bind maltose and nigerose with α 1-4-bonds. However, this requires experimental evidence.

Besides CORK1, there are 13 additional LRR-MD-RLKs in *Arabidopsis*. They have been shown to be involved in lipopolysaccharide perception [84], pollen tube development [85] and control of cell death in leaves [86]. Several of them are upregulated by brassinosteroids and participate in immune responses [87,88,89]. However, whether their MDs bind to sugars is not known. Due to structure and sequence similarity, they might interact with other sugars from cell wall polymers, since the *cork1* phenotype excludes their participation in COM responses.

Intriguingly, the phosphoproteomic study identified FER as a phosphorylation target after CT treatment (Table 2). It harbors a malectin-like domain (MLD), consisting of two tandem malectin domains. It has been shown to regulate CWI and pollen tube development, although known ligands for FER are RALF peptides and pectins [19,22,90,91]. Together with the phosphorylation of several members of CSC, it is likely that CORK1 controls cell wall repair mechanism, and it may coordinate FER upon cell wall damage.

### 4.2. Crosstalk between CORK1 and Other Signaling Pathways

Activation of *FLS2,* genes involved in FLS2 signaling and FLS2 targets, phosphorylation of MAPKs (MAPKKK3, MPK3 and MPK6), plasma membrane-localized SERK1 and FER by CT/CORK1 suggest crosstalk to other receptor kinases and PAMP-activated defense signaling (Figure 7, Table 1 and Table 2). Souza et al. [11] and Johnson et al. [13] have shown that combined treatments of CB/CT with either flg22 or chitin trigger higher [Ca^2+^]_cyt_ levels, ROS production and MPK3/6 phosphorylation. CWI signaling and plant defense are tightly coupled: both operate via [Ca^2+^]_cyt_ elevation. Although their Ca^2+^ signatures might differ, the overlap is apparent by the large number of defense-related genes that respond to CT/CORK1 activation and PAMP-induced signaling. Likewise, *WRKY30* and *WRKY40* are downstream targets of COM/CORK1 activation, and these transcription factors participate in various biotic and abiotic responses [92,93]. Understanding the crosstalk between CORK1 and other PRRs as well as their signaling components will provide a broader picture of how plants integrate different threats and developmental signals. CWI signaling is also important during many developmental processes, starting from growth, division and differentiation of cells, meristem development, senescence and to fertilization. The tissue-specific expression of the different members of the MD-containing receptor kinases might reflect their different roles in monitoring cell wall alterations [94].

### 4.3. CT Regulates Metabolism of Aromatic Amino Acids and Secondary Metabolites

Tryptophan-derived secondary metabolites have long been considered important components for innate immunity, and Trp serves as the starting amino acid for the biosynthesis of camalexin and indolic glucosinolates [95]. Upregulation of Trp biosynthesis is crucial for plant defense against fungal pathogens and hemibiotrophs [96,97,98].

Although several genes important for flagellin-induced callose deposition [99] are also upregulated by CT, callose deposition could not be observed [11]. This might be due to the low induction of *PEN2* and *PMR4* by CT (Figure 7). Interestingly, *ABCG36* (*PEN3*) was upregulated, and the protein, ABCG36/PEN3, was also phosphorylated by CT (Figure 7 and Table 2). Phosphorylation of the transporter is important for pathogen defense, and it has been proposed that it participates in the deposition of defense-related secondary metabolites [100]. Which components can be exported by ABCG36 (PEN3) are not known, but they might well be important for cell wall repair. Besides differences between COM (DAMP) and flagellin (PAMP), our -omics analyses confirm crosstalk at the signaling levels. Besides COM/CORK1-induced investment into defense, this might play an important role in priming plant immune responses induced by other stimuli.

Another group of CT-stimulated genes involved in phenylpropanoid metabolism (Figure 7) convert Phe through cinnamic acid (Phe-ammonia lyase), *p*-coumaric acid (cinnamate-4-hydroxylase) to *p*-coumaryol-CoA (4-coumarate:CoA ligase 1). From there, *p*-coumaryol-CoA can be used to synthesize lignin with cinnamyl alcohol dehydrogenase 5 (CAD5), caffeoyl-CoA O-methyltransferases and peroxidase 4 [51,52]. These steps are important in secondary cell wall synthesis.

Apart from aromatic compounds, expression of lipoxygenases involved in jasmonic acid (JA) biosynthesis are upregulated by CT (Table 1). Moreover, the gene for CML42, a negative regulator of JA signaling and biosynthesis, is downregulated by CT (Table 1) [101]. Furthermore, JOX2, which converts JA to 12-hydroxyjasmonate, an inactive form of JA [102], is phosphorylated at Ser369 in response to CT. The enzyme prevents over-accumulation of JA and its bioactive form JA-Ile under stress [103,104] and thus, represses basal JA defense responses. These results show that JA is a target of COM signaling. However, it appears that COM treatment established a moderate and balanced JA level in the cell. Comparative analysis of the regulated genes and phosphorylation targets suggests that the primary effect of COM/CORK1 signaling is to activate cellular processes that strengthen the cell wall and those promoting cytoplasmic immunity.

This work provides evidence that the LRR receptor kinase CORK1 is required for the many cellular responses induced by cellulose breakdown fragments, and demonstrates the importance of the malectin domain for COM sensing. With recent findings of other cell wall breakdown products acting as DAMPs [12,14,15,16], closer inspection of the other LRR-MD RLK members might be a reasonable strategy to identify receptors for other cell wall polysaccharide breakdown products. Our transcriptome and phosphoproteome analyses provide a list of components that are potentially involved in COM signaling, might represent COM targets, or participate in CORK1 crosstalk.

## Figures and Tables

**Figure 1 cells-11-02960-f001:**
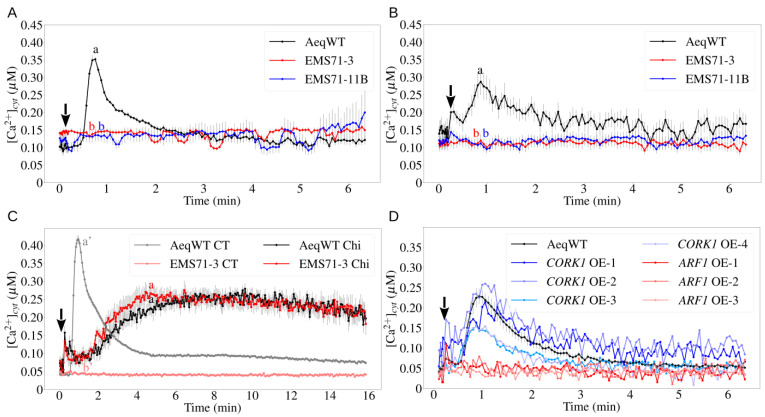
Identification of *CORK1* through EMS mutagenesis: (**A**) and (**B**) Cytoplasmic calcium elevation by 10 µM CT in root (**A**) and leaf (**B**) tissue of aequorin wild-type (AeqWT) and two independent, CT non-responsive EMS lines (EMS71) named EMS71-3 and EMS71-11B. Error bars represent SE from at least 10 seedlings. (**C**) Cytoplasmic calcium elevation by 10 µM CT or 10 µM chitohexaose (chi) in root tissue of AeqWT and EMS71-3. Error bars represent SE from 8 seedlings. (**D**) Cytoplasmic calcium elevation by 10 µM CT in leaf tissue of EMS71-3 complemented with *CORK1* (*CORK1*-OE) or *ARF1* (*ARF1*-OE). Error bars represent SE from 12 seedlings for AeqWT. Arrows indicate the onset of elicitor application. Statistical significance at the peak value was determined by Tukey’s HSD test with *p* ≤ 0.05 and is indicated by different lowercase letters. All experiments were repeated at least 3 times with similar results.

**Figure 2 cells-11-02960-f002:**
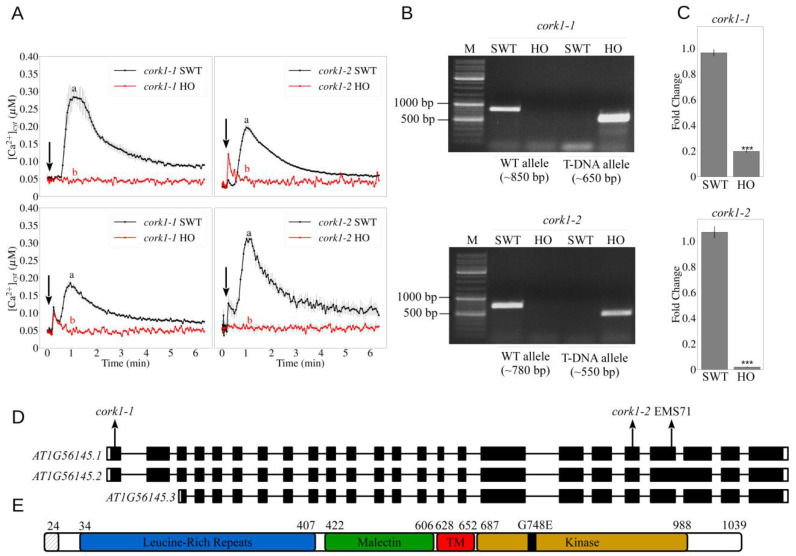
T-DNA mutants for *CORK1* do not respond to COM. (**A**) Cytoplasmic calcium elevation by 10 µM CT in root (upper panels) and leaf (bottom panels) tissues of T-DNA mutants crossed to aequorin wild-type. Error bars represent SE from at least 5 seedlings. SWT/HO: segregated wild-type/homozygous mutant from the cross to aequorin wild-type. Arrows indicate the onset of elicitor application. Statistical significance at the peak value was determined by Tukey’s HSD test with *p* ≤ 0.05 and is indicated by different lowercase letters. The experiment was repeated at least 3 times with similar results. (**B**) Genotyping of the SWT and HO seedlings. Wild-type allele is confirmed with the primer set LP and RP of the respective T-DNA insertion line. T-DNA allele is confirmed with the primer set LB_SALK and RP of the respective T-DNA insertion line. Annealing temperature for the PCR reactions is 58 °C. M: DNA marker (ladder); bp: base pair. (**C**) *CORK1* expression in root tissue of SWT and HO seedlings. Error bars represent SE from 3 independent biological replicates, each with 5 seedlings. Statistical significance was determined by Student’s T-test based on ΔCq values between the two genotypes (*** *p* ≤ 0.001). (**D**) Gene model for *CORK1* (*AT1G56145*). Two T-DNA insertion mutants used in this study are named *cork1-1* (SALK_099436C; N671776) and *cork1-2* (SALK_021490C; N674063). Position of the SNP induced by EMS mutagenesis is labeled EMS71. Arrows indicate the approximate location of T-DNA insertions and SNP on the gene. (**E**) Predicted protein structure of CORK1. Positions of amino acid residues are shown by numbers. The first 24 amino acids are predicted to be a signal peptide. G748E indicates the amino acid substitution from glycine to glutamate found in EMS71. TM: transmembrane domain.

**Figure 3 cells-11-02960-f003:**
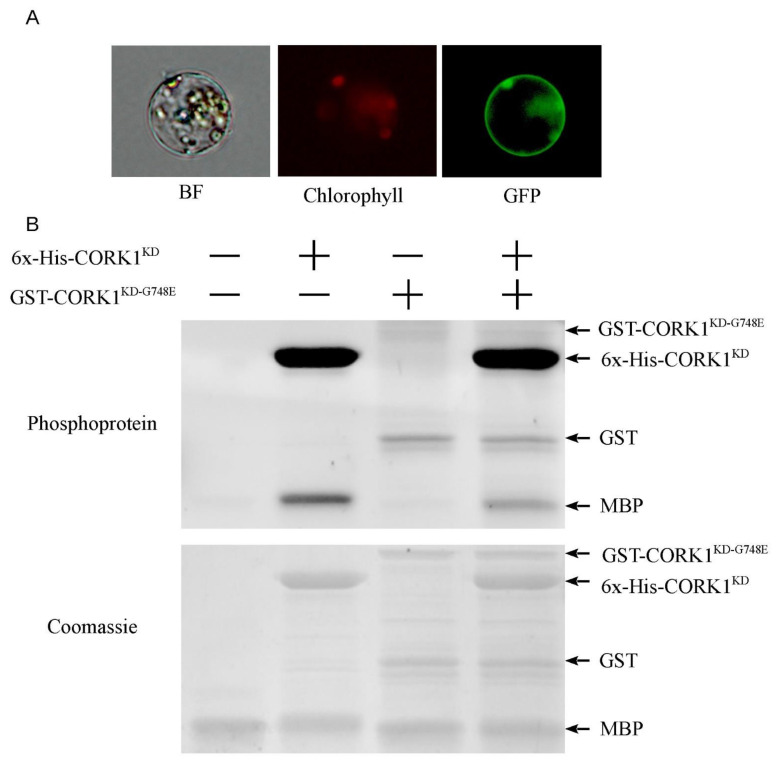
*CORK1* encodes a functional membrane-bound receptor kinase. (**A**) Subcellular localization of GFP-tagged CORK1 in *Arabidopsis* mesophyll protoplast. (**B**) Phosphorylation of the substrate MBP (myelin basic protein) by CORK1^KD^ but not by CORK1^KD-G748E^; 6x-His: polyhistidine tag with 6 histidine residues; GST: glutathione-S-transferase tag. The plus (+) and minus (−) signs indicate the presence or the absence of the expressed protein, respectively.

**Figure 4 cells-11-02960-f004:**
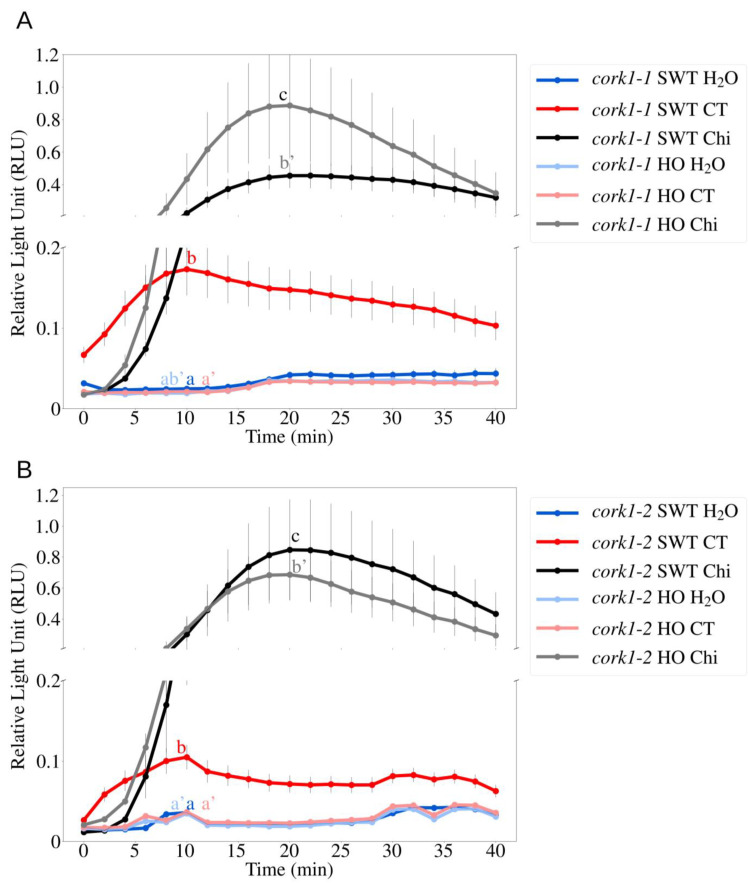
The *cork1* mutants failed to induce ROS production upon COM perception. CT (10 µM) triggers ROS production in root tissue in SWT but not in HO seedlings of (**A**) *cork1-1* and (**B**) *cork1-2*. ROS production by application of 10 µM chitohexaose (Chi) was not affected by the mutation. SWT/HO: segregated wild-type/homozygous mutant from the cross to aequorin wild-type. Error bars represent SE from at least 6 seedlings for each treatment. Statistical significance at the peak value was determined by Tukey’s HSD test with *p* ≤ 0.05 and is indicated by different lowercase letters. The experiment was repeated 3 times with similar results.

**Figure 5 cells-11-02960-f005:**
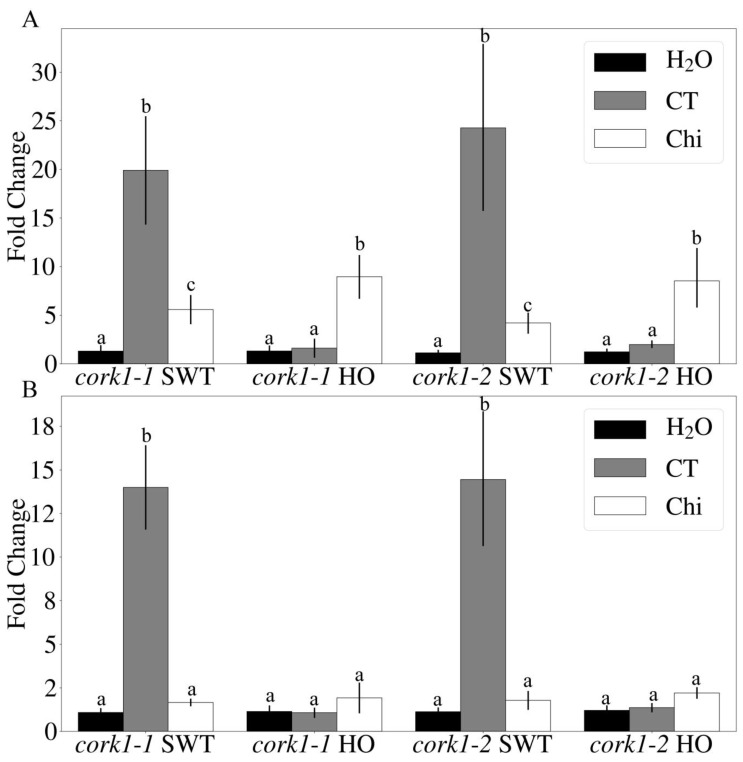
Upregulation of *WRKY30* and *WRKY40* mRNA levels in root tissue by COM is *CORK1*-dependent. (**A**) *WRKY30* and (**B**) *WRKY40* mRNA levels 1 h after 10 µM CT or 10 µM chitohexaose (Chi) treatment in *cork1-1* and *cork1-2* SWT (segregated wild-type) and HO (homozygous mutant) from the cross to aequorin wild-type. Values were normalized to water treatment on the same genotype. Error bars represent SE from at least 4 independent biological replicates, each with 16 seedlings. Statistical significance was determined by Tukey’s HSD test based on ΔCq values with *p*-value ≤ 0.05 and is indicated by different lowercase letters.

**Figure 7 cells-11-02960-f007:**
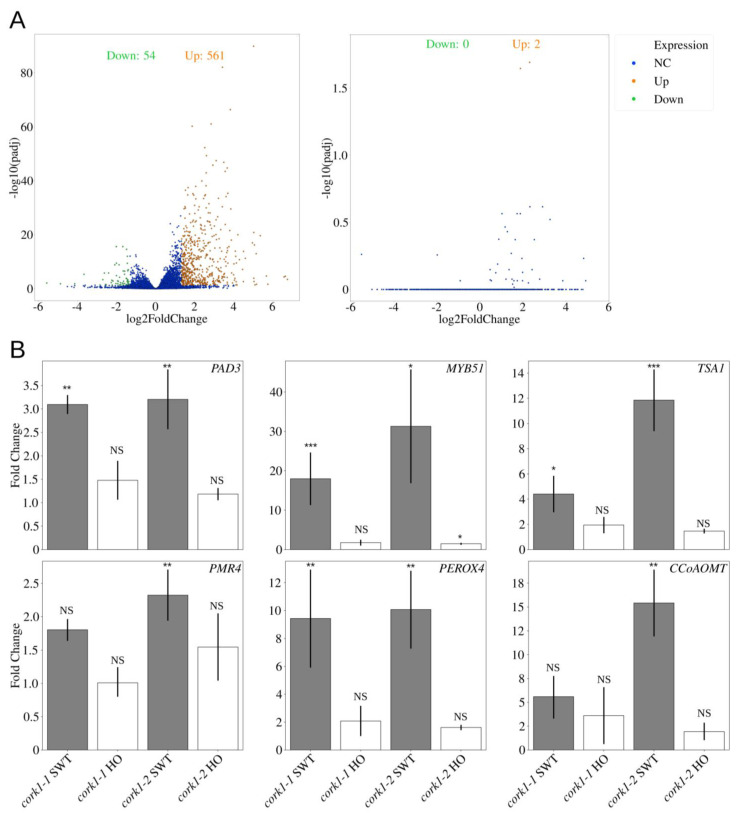
CT-regulated genes. (**A**) Volcano plots showing the distribution of differentially expressed genes in root tissue. Left: 10 µM CT treatment compared to water control in SWT. Right: 10 µM CT treatment compared to water control in HO. NC: no change; Up: upregulation; Down: downregulation; padj: adjusted *p*-value using Benjamini and Hochberg method. The FDR cutoff value is set as 0.1. (**B**) qPCR analysis of candidate genes regulated by 10 µM CT in root tissue of *cork1-2* SWT and HO seedlings. Values were normalized to water treatment on the same genotype. SWT/HO: segregated wild-type/homozygous mutant from the cross to aequorin wild-type. Error bars represent SE from 4 independent biological replicates, each with 16 seedlings. Statistical significance was determined by Student’s T-test based on ΔCq values (NS: not significant; * *p* ≤ 0.05; ** *p* ≤ 0.01; *** *p* ≤ 0.001).

**Table 1 cells-11-02960-t001:** Differentially expressed candidate genes by 10 µM CT compared to water control in root tissue of *cork1-2* segregated wild-type from the cross to aequorin wild-type.

Process	Accession No.	Annotation	log2FoldChange	padj
Tryptophan biosynthesis	*AT3G54640*	*TSA1*	2.93	1.54 × 10^−46^
	*AT5G05730*	*ASA1*	3.83	4.27 × 10^−67^
	*AT1G25220*	*ASB1*	3.43	8.09 × 10^−83^
Flagellin perception/callose deposition/wall thickening/indolic glucosinolate biosynthesis	*AT5G46330*	*FLS2*	1.93	6.18 × 10^−11^
	*AT2G19190*	*FRK1*	2.22	4.26 × 10^−15^
	*AT4G23550*	*WRKY29*	1.37	4.38 × 10^−10^
	*AT1G18570*	*MYB51*	3.67	1.76 × 10^−45^
	*AT1G24100*	*UGT74B1*	2.23	4.37 × 10^−31^
	*AT5G57220*	*CYP81F2*	3.59	8.92 × 10^−11^
	*AT4G31500*	*CYP83B1/SUR2*	2.79	8.38 × 10^−29^
	*AT1G59870*	*ABCG36*	1.46	2.00 × 10^−9^
	*AT2G44490*	*PEN2*	0.89	2.35 × 10^−7^
	*AT4G03550*	*PMR4*	0.73	2.97 × 10^−4^
Camalexin biosynthesis	*AT3G26830*	*CYP71B15/PAD3*	1.89	3.08 × 10^−4^
Jasmonic acid biosynthesis	*AT1G55020*	*LOX1*	1.87	5.91 × 10^−61^
	*AT1G17420*	*LOX3*	3.53	2.31 × 10^−6^
	*AT1G72520*	*LOX4*	2.44	1.69 × 10^−18^
Glucosinolate biosynthesis	*AT1G74100*	*SOT16*	2.52	3.37 × 10^−35^
	*AT1G18590*	*SOT17*	2.78	2.18 × 10^−25^
Phenylpropanoid metabolism/biosynthesis	*AT2G37040*	*PAL1*	1.55	2.79 × 10^−15^
	*AT2G30490*	*C4H*	1.65	1.75 × 10^−20^
	*AT1G51680*	*4CL1*	2.08	7.53 × 10^−26^
	*AT1G67980*	*CCOAOMT*	3.67	6.47 × 10^−5^
	*AT4G34230*	*CAD5*	1.60	6.77 × 10^−10^
	*AT1G14540*	*PER4*	2.44	1.93 × 10^−22^
Leaf senescence	*AT5G24110*	*WRKY30*	3.29	7.20 × 10^−16^
ABA signaling	*AT1G80840*	*WRKY40*	2.75	6.59 × 10^−9^
Plant-pathogen interaction	*AT2G19990*	*PR-1-Like*	-2.62	1.59 × 10^−6^

**Table 2 cells-11-02960-t002:** Candidate proteins with significant alteration in phosphorylation upon 10 µM CT treatment compared to water control in root tissue of *cork1-2* SWT. Numbers between brackets indicate the probability of the modification on the amino acid residue. Calculation for the corrected fold change and the ratio-adjusted *p*-value is described in the Section 2. SWT: segregated wild-type from the cross to aequorin wild-type. The field ‘Modifications in Proteins’ provides detailed information on the number of amino acid modification of the indicated protein accession number. Phospho: phosphorylation; Acetyl: acetylation; N-term: N-terminus; Met-loss: loss of methionine.

Comparison	UniProt Accession No.	Annotation	Peptide Sequence	Modifications in Proteins	Corrected Fold Change	Ratio-Adjusted *p*-Value
5 min	Q9XIE2	ABCG36	RTQSVNDDEEALK	Q9XIE2 1xPhospho [S45(100)]	10.64	1.74 × 10^−2^
	Q9XIE2	ABCG36	TQSVNDDEEALK	Q9XIE2 1xPhospho [S45(100)]	6.97	8.76 × 10^−3^
	Q9XIE2	ABCG36	NIEDIFSSGSR	Q9XIE2 1xPhospho [S40(99.4)]	4.32	9.41 × 10^−3^
	Q9XIE2	ABCG36	NIEDIFSSGSRR	Q9XIE2 1xPhospho [S40(99.7)]	2.50	4.26 × 10^−4^
	Q9FL69	AGD5	MESAATPVER	Q9FL69 1xPhospho [T206(100)]	17.18	1.18 × 10^−2^
	Q9C636	CC1	TDSEVTSLAASSPARSPR	Q9C636 2xPhospho [S16(100); S20(100)]	2.36	4.08 × 10^−2^
	Q9C636	CC1	TDSEVTSLAASSPARSPR	Q9C636 1xPhospho [S20(100)]	2.24	2.01 × 10^−2^
	F4ISU2	PICC	DIDLSFSSPTKR	F4ISU2 1xPhospho [S1274(99.6)]	5.71	3.55 × 10^−3^
	F4ISU2	PICC	SRDIDLSFSSPTK	F4ISU2 1xPhospho [S1274(100)]	3.28	8.22 × 10^−3^
	F4ISU2	PICC	DIDLSFSSPTK	F4ISU2 1xPhospho [S1274(99.7)]	2.80	1.89 × 10^−2^
	Q941L0	CESA3	RLPYSSDVNQSPNR	Q941L0 1xPhospho [S176(100)]	2.80	3.76 × 10^−3^
	Q38868	CPK9	AAAAAPGLSPK	Q38868 1xPhospho [S69(100)]	4.10	6.33 × 10^−3^
	F4IIM1	CSI1	MHDSEPPTPHSTTK	F4IIM1 1xPhospho [T37(100)]	6.28	2.56 × 10^−2^
	Q9FMM3	EXA1	VLSSPVVTQASHK	Q9FMM3 1xPhospho [S1553(99.6)]	4.87	3.21 × 10^−4^
	Q9LUM0	FAB1B	VAYPVSPALPSK	Q9LUM0 1xPhospho [S1321(100)]	15.97	4.24 × 10^−4^
	Q9SCZ4	FER	TGPTLDHTHVSTVVK	Q9SCZ4 1xPhospho [S695(99.2)]	6.69	4.92 × 10^−3^
	O64851	IQM4	FPSPYGPIPSPRPSPR	O64851 2xPhospho [S505(100); S509(100)]	921.45	2.27 × 10^−3^
	F4JVX1; O64851	IQM4	LAYMGIPSPR	F4JVX1 1xPhospho [S520(100)]; O64851 1xPhospho [S525(100)]	45.79	1.31 × 10^−5^
	Q9FFF6	JOX2	SHVESHISPR	Q9FFF6 1xPhospho [S369(100)]	4.19	2.42 × 10^−4^
	F4HRJ4	MAPKKK3	VASTSLPK	F4HRJ4 1xPhospho [T/S]	2.32	1.68 × 10^−2^
	B3H653	MPK3	EATNLIPSPR	B3H653 1xPhospho [S16(100)]	17.07	1.53 × 10^−2^
	Q39026	MPK6	VTSESDFMTEYVVTR	Q39026 1xPhospho [Y223(100)]	239.27	4.67 × 10^−3^
	Q39026	MPK6	VTSESDFMTEYVVTR	Q39026 1xPhospho [T221(100)]	29.15	3.22 × 10^−3^
	P28187	RABA5C	QLNSDSYKEELTVNR	P28187 1xPhospho [S186(100)]	2.21	2.47 × 10^−2^
	Q9FIJ0	RBOHD	ILSQMLSQK	Q9FIJ0 1xPhospho [S347(100)]	−29.59	3.20 × 10^−3^
	Q94F62; Q94AG2	SERK1	DTHVTTAVR	Q94F62 1xPhospho [T450(99.5)]; Q94AG2 1xPhospho [T463(99.5)]	15.22	8.50 × 10^−4^
	Q39233	SYP21	MSFQDLEAGTRSPAPNR	Q39233 1xMet-loss+Acetyl [N-Term]; 1xPhospho [S12(99.2)]	65.82	1.67 × 10^−4^
	A8MQL1	TRAF1B	STAVLSSPR	A8MQL1 1xPhospho [S716(100)]	9.72	1.53 × 10^−4^
	F4KHU8	UXS3	QNTTKPPPSPSPLR	F4KHU8 1xPhospho [S31(100)]	3.13	4.96 × 10^−3^
	Q39160	XI-1	AGATGSITTPR	Q39160 1xPhospho [T1195(100)]	5.97	2.51 × 10^−2^
15 min	Q9FL69	AGD5	MESAATPVER	Q9FL69 1xPhospho [T206(100)]	16.41	6.12 × 10^−5^
	F4ISU2	PICC	DIDLSFSSPTKR	F4ISU2 1xPhospho [S1274(99.6)]	5.71	3.55 × 10^−3^
	F4ISU2	PICC	DIDLSFSSPTKR	F4ISU2 1xPhospho [S1274(99.6)]	5.40	4.15 × 10^−2^
	F4ISU2	PICC	DIDLSFSSPTK	F4ISU2 1xPhospho [S1274(99.7)]	4.42	2.98 × 10^−2^
	Q38868	CPK9	AAAAAPGLSPK	Q38868 1xPhospho [S69(100)]	5.02	1.51 × 10^−2^
	Q9FHK4	EDR4	SLQLEGPGGR	Q9FHK4 1xPhospho [S322(100)]	5.44	7.39 × 10^−3^
	Q9FMM3	EXA1	MTTSSHPPPSPVPTTQK	Q9FMM3 1xPhospho [S1449(100)]	43.54	1.45 × 10^−2^
	F4JVX1; O64851	IQM4	LAYMGIPSPR	F4JVX1 1xPhospho [S520(100)]; O64851 1xPhospho [S525(100)]	844.25	6.30 × 10^−5^
	F4JVX1; O64851	IQM4	LAYMGIPSPR	F4JVX1 1xPhospho [S520(100)]; O64851 1xPhospho [S525(100)]	99.10	1.57 × 10^−3^
	O64851	IQM4	FPSPYGPIPSPRPSPR	O64851 2xPhospho [S505(100); S509(100)]	355.82	3.33 × 10^−5^
	Q84M93	MPK17	LEEHNDDEEEHNSPPHQR	Q84M93 1xPhospho [S397(100)]	51.33	5.67 × 10^−3^
	B3H653	MPK3	EATNLIPSPR	B3H653 1xPhospho [S16(100)]	196.12	3.60 × 10^−2^
	Q39026	MPK6	VTSESDFMTEYVVTR	Q39026 1xPhospho [T221(100)]	33.63	6.00 × 10^−4^
	Q9LM33	MPK8	HHASLPR	Q9LM33 1xPhospho [S495(100)]	−3.12	8.53 × 10^−3^
	Q9LRP1	NPSN13	ELKDEEARNSPEVNK	Q9LRP1 1xPhospho [S74(100)]	−184.99	1.13 × 10^−2^
	Q93YU5	SEC8	ASQHDINTPR	Q93YU5 1xPhospho [T482(100)]	88.63	1.71 × 10^−2^
	A8MQL1	TRAF1B	STAVLSSPR	A8MQL1 1xPhospho [S716(100)]	−3.01	2.55 × 10^−2^
	F4KHU8	UXS3	QNTTKPPPSPSPLR	F4KHU8 1xPhospho [S31(100)]	4.48	1.22 × 10^−2^
	Q9SN95	UXS5	QTSPKPPPSPSPLR	Q9SN95 2xPhospho [S15(100); T/S]	3.13	1.18 × 10^−2^
	P93043	VPS41	REDNNRSSFSQR	P93043 1xPhospho [S860(99.7)]	351.09	7.54 × 10^−3^
	Q96289	ZAT10	MALEALTSPR	Q96289 1xMet-loss+Acetyl [N-Term]; 1xPhospho [S8(100)]	2765.33	1.77 × 10^−4^

## Data Availability

Raw sequences for the GWAS and the transcriptome analysis have been deposited to the NCBI Gene Expression Omnibus (GEO) database (https://www.ncbi.nlm.nih.gov/geo/) with the accession no. GSE197891 and GSE198092, respectively. The mass spectrometry proteomics data have been deposited to the ProteomeXchange Consortium via the PRIDE partner repository (http://www.ebi.ac.uk/pride) with dataset identifier PXD033224.

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
