# Peer review of "CORK1, A LRR-Malectin Receptor Kinase, Is Required for Cellooligomer-Induced Responses in Arabidopsis thaliana"

_cells, 2022, doi:10.3390/cells11192960_

Round 1

Reviewer 1 Report

Evidently this topic is of interest to the Plant Biology community. The experiments are well designed, the methodology adequately described, and the results well presented and discussed. Some minor drafting errors should be corrected, mainly regarding the definition of acronyms that, in my opinion, their origin should be indicated with capital letters, as well as avoiding the use of a letter s at the end to indicate plural.

Another thing that should be corrected is to use ≤ instead of just <, as was done in all cases.

All my suggestions in this regard are highlighted in the attached document.In particular, my suggestions that the presentation of the results should be improved and that the support of the conclusions could be improved, come from the fact that Tables 1 and 2 were not included in the manuscript, while two sections of results (those referring to the transcriptome and phosphoproteome) make constant reference to the data apparently compiled in these Tables.

Author Response

Reviewer 1:

Comments and Suggestions for Authors:

Evidently this topic is of interest to the Plant Biology community. The experiments are well designed, the methodology adequately described, and the results well presented and discussed. Some minor drafting errors should be corrected, mainly regarding the definition of acronyms that, in my opinion, their origin should be indicated with capital letters, as well as avoiding the use of a letter s at the end to indicate plural.

Another thing that should be corrected is to use ≤ instead of just <, as was done in all cases.

All my suggestions in this regard are highlighted in the attached document.In particular, my suggestions that the presentation of the results should be improved and that the support of the conclusions could be improved, come from the fact that Tables 1 and 2 were not included in the manuscript, while two sections of results (those referring to the transcriptome and phosphoproteome) make constant reference to the data apparently compiled in these Tables.

-Thank you for pointing out the issue. We have modified the text according to your detailed suggestions and included Table 1 and Table 2 which contains organized information on the supplementary datasets, and genes/proteins described in the text. From all data presented in the Suppl. Material those relevant for this paper were extracted, information added and presented in Table 2. In addition, additional changes in response to Reviewer 2 partially overlapped with your critical points and they were changed as well. We hope that these corrections take care of your concerns, and that the manuscript is now in a better shape.

Reviewer 2 Report

This is a detailed and carefully executed study which significantly enhances knowledge of cellooligomer-induced responses in plant, by adding a new and functionally significant protein, CORK1. In this study, the author started from the mutagenesis screening to identify regulators that are involved in COM perception. The author identified CORK1, a malectin domain containing leucine-rich repeat transmembrane protein kinase. CORK1 is required for response to COM and loss of CORK1 in Arabidopsis failed to respond to COM and cannot produce ROS and induce calcium elevation, suggesting that CORK1 is an important player for CT response. RNA-seq and phosphoproteome further confirmed that CORK1 is involved in cell wall surveillance, secondary metabolite and maintenance of defense response. Overall, I found the data to be compelling, claims appropriately backed with data. For these reasons, I strongly recommend considering this manuscript for publication in Cells with the following minor revisions.

For the last result part: COM/CORK1-mediated changes in the phosphoproteome pattern in roots. From the supplementary dataset2, it’s hard for readers to know detailed info of these phosphorylated proteins. Those data list should be regenerated and include candidate ID, name, functions, et. The author did not validate the phosphoproteome data and it is not accurate to claim that these proteins can be phosphorylated by COM/COCK1, such as CESA1 and CSI1 (l563-564). The author needs either to carefully explain them or to confirm those proteins are indeed substrates of COCK1 protein. On the other hand, the author needs to reorganize and simplify this part. For example, the author explained that how some kinases are involved in endomembrane system and secretory pathway (l570-580), which seems to distract readers’ attention from the main results of the text.  

Table 2 is missing from the manuscript.

Line73 “CORK1 mutants” should be “cork1 mutants”

Line124: “described,” needs a citation or changes to “described in the below section”.

Line131 and line 133: “adjusted p-value < 0.05” and “default FDR cutoff was set as 0.1” are conflicted. Please claim them clearly.

Line210: “fold change of >2, ratio-adjusted pvalue <0.05 (pvalue/log4ratio)” should be “fold change of ratio > 2, adjusted p-value < 0.05”. What’s the meaning of pvalue/log4ratio? Could the author explain it clearly?

Line364 “EMS71” bold words. The same to Line576; Line712; the reference.

Line383-384: “the exon located near the 3’end” It would be better to indicate the exact exon that the T-DNA inserted in.

Line386: “latter mutation” can be changed to “single mutation”.

Line 407 “cork1-1”, “cork1-2” italicize. Please double check the same little errors from the whole text.

Line 416-line417: “The GFP signal at the plasma membrane confirmed that CORK1 is a mem-416 brane-associated protein”. The author confirmed CORK1 is a membrane-associated protein showing in Fig. 3A. autofluorescence needs to be changed with chlorophyll. Also, there is partial colocalization of CORK1-GFP and chlorophyll. The author should provide a good quality imaging of CORK1-GFP. Usually, PM-labels such as FM-4-64 is used as a good marker. It’s better to show the colocalization of FM-4-64 and CORK1-GFP to confirm the membrane association of CORK1. 

Line 420-421: “Next, the cytoplasmic region encompassing the kinase domain (CORK1KD) was 420 cloned into the expression vector pET28a”. It’s not necessary to explain kinase domain was cloned into pET28a that should be edited simply in the text. CORK1KD was cloned into pET28a, which contains 6XHis. All the recombinant protein of CORK1KD-6XHis or 6xHis-CORK1KD were lacking from the text. The author needs to clarify CORK1KD in the text and figures, like GST-CORK1KD-G748E. Moreover, from figure 3B, how the author defined the upper band is GST-CORK1KD-G748E. Protein band of GST-CORK1 KD-G748E indicated by the arrow is so faint and another specific band is just below the arrow. A western blot to display these tagged proteins would be better to confirm these recombinant proteins.

Line458: “lack of the calcium elevation was not due to the absence of the GFP fusion protein, as GFP 458 signal was present at the plasma membrane (Fig. 6C)”. The main point is the lack of the calcium elevation was not due to the localization change or the absence of ROCK1-GFP protein. The membrane labeled in the figure 6C should be clarified clearly.

Line 468: extra dot.

Line 487: “target”. CORK1 is a membrane-bound receptor like kinase, and it would be better to use “responsive”.

Line487 and l540 and l565: “COM/CORK” should be “COM/CORK1”.

Author Response

Dear Reviewer:

                Thank you very much for your detailed suggestions on the manuscript. We have modified the text and the figures according to your suggestions. Table 1 and Table 2 are now included into the manuscript and they should take care of one of your major concerns. We also provided explanations regarding your questions on the manuscript. Please see the answer below each points. Thank you again for your valuable advice.

Round 2

Reviewer 1 Report

I appreciate that you have listened to my suggestions, your research is very interesting, the experiments are properly designed and executed, the results well presented and the conclusions adequately supported. Congratulations.

Author Response

Dear Reviewer,

Thank you very much for your comments and suggestions for our manuscript.